# Atomic-scale visualization of the interlayer Rydberg exciton complex in moiré heterostructures

Meng Zhao[1,2,9], Zhongjie Wang [1,2,9] ✉, Lu Liu [1,2,3,9], Chunzheng Wang[1,2], Cheng-Yen Liu [1,2], Fang Yang [4,5], Hua Wu [1,2,3,6] ✉ & Chunlei Gao[1,2,4,5,6,7,8] ✉

Excitonic systems, facilitated by optical pumping, electrostatic gating or magnetic field, sustain composite particles with fascinating physics. Although various intriguing excitonic phases have been revealed via global measurements, the atomic-scale accessibility towards excitons has yet to be established. Here, we realize the ground-state interlayer exciton complexes through the intrinsic charge transfer in monolayer $YbCl_3$/graphite heterostructure. Combining scanning tunneling microscope and theoretical calculations, the excitonic in-gap states are directly profiled. The out-of-plane excitonic charge clouds exhibit oscillating Rydberg nodal structure, while their in-plane arrangements are determined by moiré periodicity. Exploiting the tunneling probe to reflect the shape of charge clouds, we reveal the principal quantum number hierarchy of Rydberg series, which points to an excitonic energy-level configuration with unusually large binding energy. Our results demonstrate the feasibility of mapping out the charge clouds of excitons microscopically and pave a brand-new way to directly investigate the nanoscale order of exotic correlated phases.

Coulomb interaction between electrons and holes leads to the formation of exciton complex[1], which are tightly bound state of multiple particles including exciton, trion[2,3], biexciton[4], and aggregate with more components[5,6]. The intricate many-component degrees of freedom inside exciton complex could be elusive, but the few-body wavefunctions generally manifest as "fat" Rydberg orbitals with large spatial extension[7,8], akin to the positronium and atoms of small nuclear charge number. Moreover, the exciton complex ensemble exhibits exotic bosonic or fermionic many-body phenomena under the description of Boson/Fermi-Hubbard model[9–13]. In particular, the interlayer excitons with separated electrons ($e$) and holes ($h$) residing in adjacent layers respectively, experience strong mutual dipole-dipole

repulsion and moiré periodic potential provided by interlayer stacking, constituting a rich platform where exciton superfluidity and extensive moiré-dominated exotic phases have been reported[14–24].

Various approaches have been used to maintain long-lived excitons, and to characterize their holistic property and phases. The former includes the usage of negative or small band gap in exciton insulator[19,25,26], application of magnetic field in atomic bilayer[12,15,24], optical excitation[22,27] and electrostatic gating[18,21,23,28,29] in the heterostructure with type II band alignment, while the latter includes ensemble-averaged techniques such as optical detection[17,21–23], capacitance measurements[9,28], compressibility characterizations[11,14] and Coulomb-drag/Counter-flow transport[12,15].

[1]State Key Laboratory of Surface Physics and Department of Physics, Fudan University, Shanghai 200438, China. [2]Shanghai Qi Zhi Institute, Shanghai 200232, China. [3]Laboratory for Computational Physical Sciences (MOE), Fudan University, Shanghai 200438, China. [4]Institute for Nanoelectronic Devices and Quantum Computing, Fudan University, Songhu Rd. 2005, Shanghai 200438, China. [5]Zhangjiang Fudan International Innovation Center, Fudan University, Shanghai 201210, China. [6]Collaborative Innovation Center of Advanced Microstructures, Nanjing University, Nanjing 210093, China. [7]Shanghai Research Center for Quantum Sciences, Shanghai 201315, China. [8]Shanghai Branch, Hefei National Laboratory, Shanghai 201315, China. [9]These authors contributed equally: Meng Zhao, Zhongjie Wang, Lu Liu. ✉e-mail: zhongjiewang18@fudan.edu.cn; wuh@fudan.edu.cn; clgao@fudan.edu.cn

In this work, we propose new strategies to realize ground-state interlayer excitons in moiré heterostructure, and to visualize them at atomic-scale. First, a huge work function difference between disparate materials can be utilized to drive a considerable charge transfer across the interface, which naturally gives rise to spontaneous ground-state interlayer excitons[30,31]. Second, by employing in-depth STM measurements and analysis, we demonstrate the capability of tunneling probe to directly profile the charge clouds of interlayer excitonic states. Concretely, we fabricate the monolayer $YbCl_3$/HOPG heterostructure via molecular beam epitaxy, and present systematic STM investigation on the interlayer excitonic states featuring Rydberg-like fingerprints. Owing to the extreme spatial and energy resolution provided by tunneling probe, we reveal the excitonic energy diagram, and map out the three-dimensional distributions of Rydberg electron- and hole-states including both the out-of-plane nodal structure and moiré-dominated in-plane periodic arrangements.

## Results and discussion

### Monolayer $YbCl_3$ on HOPG and the electronic structure

The rare-earth halide $YbCl_3$ is a representative strongly correlated van der Waals material of 4f electrons[32]. Figure 1a illustrates its atomic structure where the honeycomb layer of Yb-ions is sandwiched between two layers of triangularly arranged Cl-ions. Single layer $YbCl_3$ is epitaxially grown on HOPG substrate (see Methods) with high quality as shown by STM images in Fig. 1b, c, presenting a monolayer island height of ~ 4 Å and the lattice parameter of $6.65 \pm 0.1$ Å.

Figure 2a displays the representative dI/dV spectrum (blue curve) for $YbCl_3$ monolayer taken at a relatively large tip-sample separation controlled by the setup condition of large tunneling resistance, showing electronic structure consistent with the density functional theory (DFT) computed density of states (DOS) of sole $YbCl_3$ monolayer: the $YbCl_3$ monolayer exhibits an insulating nature evinced by the nearly extinguished DOS inside the gap ranging from −0.8 eV to 0.8 eV, with the conduction band and valence band derived from Yb-4f orbitals and Cl-3p orbitals, respectively [dashed curves in Fig. 2a]. Full results of the DFT computed $YbCl_3$ DOS are displayed in Supplementary Fig. 1.

Remarkably, when dipping the tip closer to the surface, unexpected dI/dV intensity arises within the $YbCl_3$ band gap as shown in Fig. 2b, which is coined as in-gap states hereafter. As the tip approaches the surface step by step, the in-gap states gradually appear and develop in the tunneling spectrum, indicating that the in-gap states are spatially closer to the interface than the intrinsic band states of $YbCl_3$. The shape details of in-gap dI/dV spectra are highly changeable, depending on setup condition and specific tip apex on account of the tunneling mechanism explained later (see more data in Supplementary Fig. 2). However, in general, all the dI/dV curves of in-gap states exhibit

several humps superimposed on a parabola background. Some of the parabolic conductance might be contributed by the V-shape DOS of graphene broadened by interfacial imperfection, but the simple possibility that the electrons purely tunnel from the tip to HOPG substrate through the $YbCl_3$ insulating barrier is inconsistent with the hump features[33,34].

### Moiré in-gap states with orbitals of large spatial extent

By examining the three-dimensional spatial arrangement of in-gap states, it is confirmed that they are not intrinsic physical properties of sole HOPG or $YbCl_3$, but stem from the $YbCl_3$/HOPG interface. Figure 2c, d is two STM images taken at exactly the same sample area but at different bias voltages. Different orientational domains of $YbCl_3$ monolayer are naturally introduced during the growth, and form a slit at the domain boundary. The chosen sample area is representative with two neighboring domains separated by a slit. Sharing the common underneath graphene layer, these two domains differing in orientation naturally feature distinct moiré landscapes. Figure 2c taken at 1.1 V presents a regular surface morphology where two domains exhibit the same height. Distinguishingly, in Fig. 2d taken at 0.7 V, two differing superlattices are discerned, and a conspicuous height difference of tens to hundreds of picometers develops for two twist-angle domains (see Supplementary Fig. 3 for more results). According to Tersoff-Hamann theory, the constant-current image approximately reflects the iso-surface of the local density of states (LDOS) integral from the Fermi level to the imaging bias[35], that is, the integrating energy window $E_U$. Consequently, upon reducing the bias voltage below the band minimum of 0.8 V, only in-gap states within $E_U$ are involved in the tunneling process as all the intrinsic states of $YbCl_3$ conduction band are out of $E_U$ and, thus, excluded. Therefore, the remarkable height difference dominated by domain orientation shown in Fig. 2d indicates that the z-direction penetration towards the vacuum or the spectral weight of in-gap states is decisively determined by the moiré structure, proving that in-gap states are indeed outcomes of the interface.

Another exceptional phenomenon of in-gap states can be spotted in Fig. 3a, wherein the height contrast between two orientational domains can even be reversed as the tip apex being changed by microscopic perturbation. Akin to Fig. 2c, d, these two domains exhibit different lattice orientations, and regular morphology at a high bias (Supplementary Fig. 4). After reducing the bias to 0.7 V, at first, the STM-obtained height of the left domain is 120 pm lower than the right one in Fig. 3a. Interestingly, in the second-half image the height difference suddenly changes from −120 pm (the left versus the right domain) to 80 pm as soon as the tip apex undergoes a change. In terms of the uninterrupted atomic registry and middle slit (indicated by the purple dotted line), the tip change recorded in Fig. 3a implies a slight

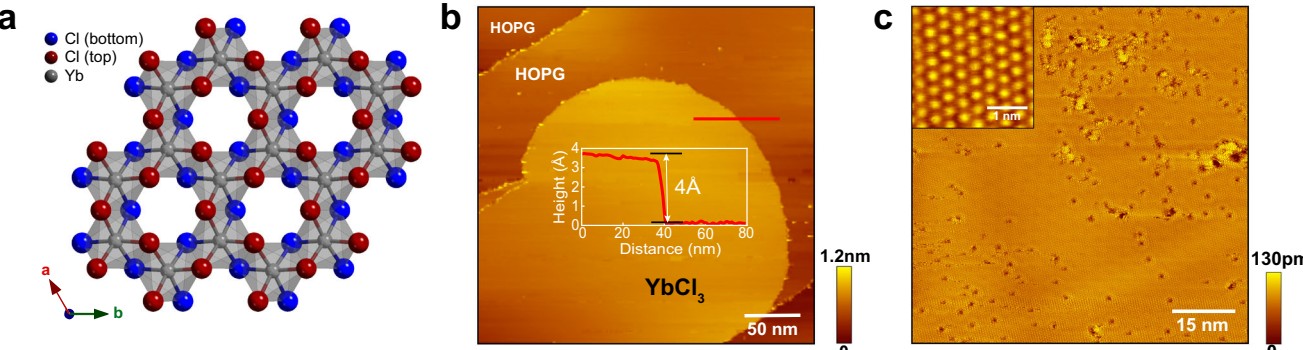

**Fig. 1 | STM images of $YbCl_3$/HOPG heterostructure. a** Schematic illustration of the crystal structure for monolayer $YbCl_3$. STM images for monolayer $YbCl_3$ on HOPG. Setup condition: **b** $U = 3.4$ V, $I = 30$ pA, **c** $U = 1.2$ V, $I = 100$ pA. The inset in (**b**) displays the height profile along the red line. The inset in (**c**) shows a close-up with atomic resolution, upon which the topmost Cl atoms are clearly resolved (setup condition for the inset: $U = 2.0$ V, $I = 100$ pA).

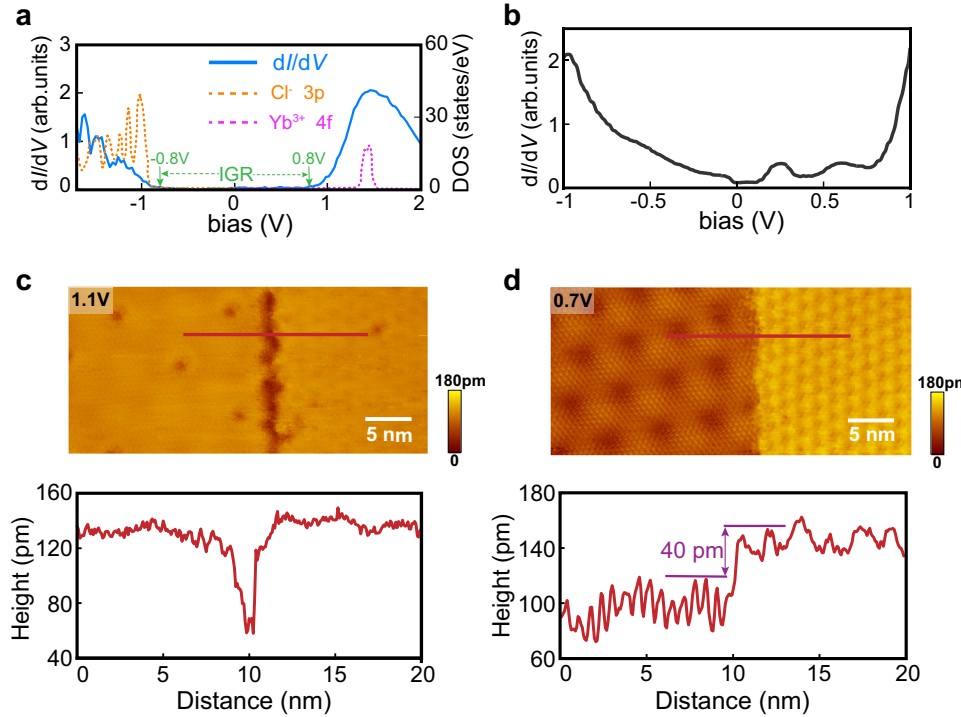

**Fig. 2 | Electronic structure and moiré in-gap states. a** Typical tunneling spectrum measured on YbCl$_3$ monolayer. The blue curve shows the differential conductance spectrum obtained at a large tip-sample distance (setup condition: $U = 2.1$ V, $I = 100$ pA). The full gap is labeled as in-gap range (IGR). The dashed curves show the computed density of states of YbCl$_3$ monolayer, the Yb 4f- and Cl 3p-bands are represented by the violet and orange dashed lines respectively. **b** Tunneling spectrum obtained at a small tip-sample distance (setup condition:

$U = 1.0$ V, $I = 100$ pA), where non-zero in-gap conductance arises. **c**, **d** STM images and height profiles for a domain boundary. The two images are taken at the same position but at different biases. The bottom panels show the corresponding height profiles along the horizontal red lines in the top panels, respectively. Setup condition: $U = 1.1$ V and $I = 10$ pA for (**c**), $U = 0.7$ V and $I = 10$ pA for **d**. Compared to (**c**) where two domains exhibit the same height, a noticeable height difference develops in (**d**).

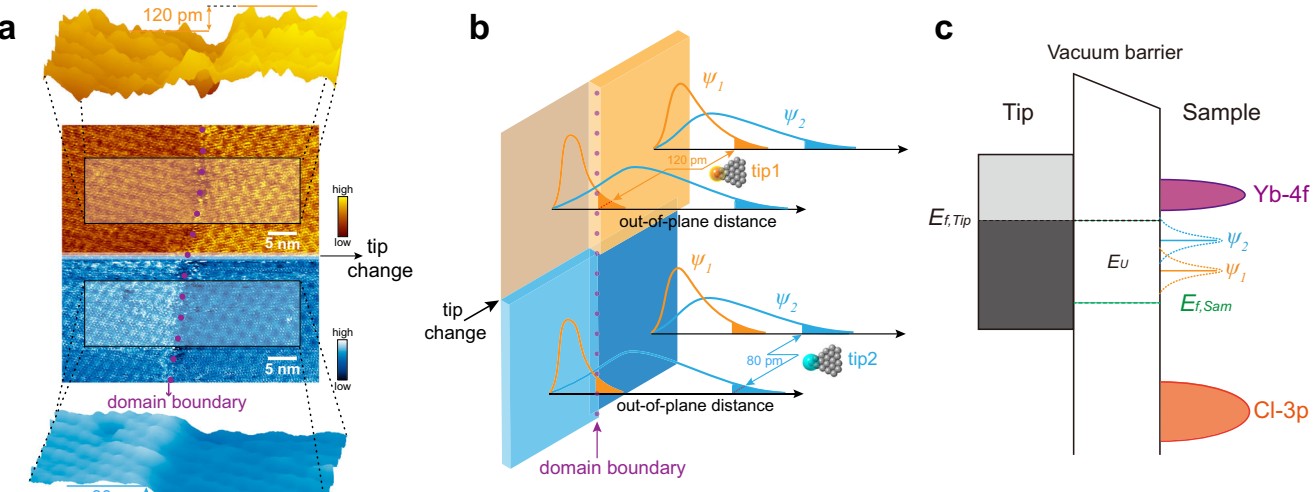

**Fig. 3 | Drastic morphological height reversal presented by moiré in-gap states. a** STM topography taken at $U = 0.7$ V, $I = 100$ pA for neighboring domains, where the in-gap states exhibit uncommon reversal of height contrast between two domains under the change of tip apex. The middle panel is the 2D STM image showing that a sudden tip change occurs at the midway point during the scanning, the images before and after the tip change are adapted to earthy yellow and cyan-blue colormap, respectively (the slit separating two domains remain uninterrupted and is indicated by the purple dotted line). The top and bottom panels are 3D plots extracted from corresponding areas before and after the tip change, respectively,

illustrating that the height contrast reverses from −120 pm (marked in the top panel) to 80 pm (marked in the bottom panel). **b** Cartoon explaining the reversal of STM-resolved height contrast, see detailed explanation in the text. **c** Energy diagram illustrating two distinguishing orbitals located at the in-gap range, and the dashed curves represent the Lorentzian broadening of in-gap levels around their eigenenergies. The bias voltage creates an energy difference between the Fermi levels of the tip and the sample, spanning a tunneling window $E_U$ which involves certain in-gap levels.

modification of the tunneling orbitals of apex atom, instead of a drastic transformation which would completely alter the apex morphology.

This is an unusual discovery in STM experiments that the electronic states originating from spatially levelled atomic planes can exhibit reversible height contrast under the modification of tip tunneling states, which involves sophisticated physics. Nevertheless, some important conclusions can already be drawn unequivocally. Firstly, the alteration of the tunneling matrix and selection rule, determined by the tunneling tip orbitals before and after the sudden change, plays a central role in producing the observed reversal of height contrast. Secondly, the in-gap states serving as tunneling destinations must contain more than one on-site quantum state with distinct orbitals, otherwise the height contrast determined by the individual state with settled spatial distribution among moiré domains could never be reversed no matter what kind of modification occurs to the tip condition. Thirdly, the multiple orbitals constituting in-gap states are significantly modulated by the moiré structure, including their evanescent wavefunction spreading into the vacuum and electronic spectral function. Fourthly, bearing in mind that the constant-current trajectory of tip roughly reflects the LDOS iso-surface of particular in-gap orbitals, the large variation of height contrast of ~ 2 Å indicates a very large spatial extent of in-gap charge clouds.

Therefore tentatively, we propose that the overall analysis leads to a scenario depicted in Fig. 3b. There are at least two largely spreading orbitals (labeled as $|\psi_1\rangle$ and $|\psi_2\rangle$, respectively) with different spatial extensions, and their wavefunctions are strongly modulated by moiré periodicity, resulting in the distinct wavefunction landscape of two orientational domains (illustrated in Fig. 3b, the wavefunctions differ in spatial distribution for left and right domains). The tip state can be expressed by the series expansion of spherical harmonic functions in terms of the $s$, $p$, $d$, ... components. Induced by the possible displacement and geometrical crystal field distortion of apex atom, the components of tip state could be altered, resulting in different tunneling matrixes with respect to the sample wavefunction before and after the tip change. If the tip orbital is more sensitive to $|\psi_1\rangle$, that is, the tunneling matrix $M_1 \equiv \langle \psi_1 | U_T | \psi_t \rangle >> M_2 \equiv \langle \psi_2 | U_T | \psi_t \rangle$ where $U_T$ is the tip potential and $\psi_t$ is tip wavefunction, the iso-current contour mainly contributed by $|\psi_1\rangle$ dominates the height difference, as the orange double-arrowed zigzag line in Fig. 3b indicates the tip trajectory across domain boundary. Once the tip tunneling state is modified into an orbital with $M_2 >> M_1$, as indicated by the cyan double-arrowed zigzag line, $|\psi_2\rangle$ will dominate the iso-current contour, leading to the reversal of height contrast.

As a result, the significant change in height contrast of ~2 Å suggests a substantial spatial expansion of in-gap orbitals on the scale of several angstroms or nanometers. Considering that the mean radius of atomic orbital is inversely proportional to the nuclear charge number $Z$, the large wavefunction extent represents hydrogenic-like orbitals with very small $Z$, which is inconsistent with the tremendous $Z$ of Yb and Cl ions. Moreover, as depicted by the energy spectrum schematic in Fig. 3c, the presence of multiple distinct orbitals within the in-gap range cannot be explained solely by the inherent band structure of the bare constituent materials in the heterostructure, neither by the forbidden band of YbCl₃ monolayer nor by the single $p_z$ orbital character of graphene Dirac cone spanning the in-gap energy range. Hitherto, the above analysis guides us to the conjecture that the in-gap states are large-orbital Rydberg series formed by a few elementary charges emerging from the interface. Next, we will prove this point by capturing the Rydberg fingerprints.

## Charge-transfer exciton complex and their Rydberg nature

To figure out the nature of the non-trivial in-gap states, we now turn to the fact that the contact of these two materials naturally results in a charge transfer interface. The work function of YbCl₃ monolayer is calculated to be 7.1 eV (the energy difference between the vacuum level and chemical potential, for an insulator/semiconductor like YbCl₃, the chemical potential is located at the middle of band gap at 0 K), which is much larger than that of 4.6 eV for graphene, driving substantial electrons to transfer from graphene underlayer to YbCl₃ monolayer and leaving transferred electrons and holes in YbCl₃ monolayer and graphene layer, respectively [Fig. 4a]. Employing Bader charge analysis[36] we theoretically estimated the density of transferred charge across the interface to be 0.21 $e$ $(h)$/nm² (see details in Methods).

The transferred $e$-$h$ states provide a possible explanation of the experimentally discovered interfacial in-gap states because the mutual Coulomb attraction binds $e$ and $h$ together into excitonic Rydberg series at the interface, with energy levels located inside the YbCl₃ gap owing to the binding energy offset[30,31]. Figure 4c shows a typical schematic diagram of $e$- and $h$- h-levels at this charge transfer interface. Considering the vanishingly small DOS at the Fermi level for the semimetal graphene, we simplify its electronic structure as a donor-type semiconductor. The principal quantum number $n = 1$ energy position of transferred $e$- ($h$-) state is lower (higher) than the conduction band minimum of YbCl₃ (valence band maximum of graphene) by a binding energy $E_B$ ($E'_B$). While Fig. 4c demonstrates the small binding energy case, Fig. 4d demonstrates a large binding energy case where some $e$-levels are lower than $h$-levels in energy. For both cases, in increasing order of energy, the principal quantum number $n$ of $e$-series is ascending while that of $h$-series is descending. It has been demonstrated in the STM studies of image potential states that the Rydberg $e$-levels form $e$-$h$ bound states are probed and manifested as discrete tunneling channels[37].

Besides the energy-level diagram of charge-transfer excitons, their few-body wavefunctions near the interface are rather complicated as the transferred $e$ and $h$ mutually bind and screen each other in an interactive manner. Especially considering the 4f orbital character inherited by the transferred $e$, the $e$-states should be highly localized. However, subjected to an almost charge-neutral lattice background, the doped electron filled into the Yb-4f band feels the strong screening of Yb ionic potential by the initial Yb-4f[13] valence electrons, rendering the Coulomb attraction from adjacent doped $h$ a comparatively dominant interaction and a large Rydberg wavefunction of transferred $e$, which is reminiscent of the shallow donors in semiconductor with large envelope wavefunction and Rydberg series[38]. Therefore, insights about the real-space wavefunction and charge cloud can be gained from a symmetry viewpoint by roughly modeling the system as electrons (or holes) trapped in one-dimensional Coulomb potential provided by the counterpart charges. Following this route, we simulated the wavefunction and probability amplitude of one-dimensional Rydberg series (see details in Methods). As shown in Fig. 4b, the charge clouds of Rydberg series permeate into the vacuum with spatially undulated nodal structure.

Next, we demonstrate that the out-of-plane nodal structure of Rydberg charge densities of $e$- and $h$-states can be profiled by the tunneling probe. As shown in the inset of Fig. 5a, detached from the tip bulk states spatially and energetically[39], the major tunneling state affixed to the apex atom is the highly localized dangling bond with dominant $d_{3z^2-r^2}$ character and slight $p_z$ character[39,40], thus providing a sufficient vertical resolution of sub-atom scale to depict the Rydberg charge clouds via current-height ($I$–$z$) characteristics (see Methods). Referring to Chen's rule and the Rydberg series provided in Fig. 4b, we derive the numerical simulation of current evolution as a function of tip-interface distance regarding tip p- and d- orbital (denoted as $I^n_{d_{3z^2-r^2}}(z)$ and $I^n_{p_z}(z)$, see details in Methods). The results are plotted in Fig. 5a, b, we can see that the $I$–$z$ characteristic remains the nodal structure as the Rydberg series does (Fig. 4b), keeping the trend that the larger the $n$ is, the more nodes the curve presents. The analytical results have a good consistency compared to the experimental curve in Fig. 5c taken at 0.7 V, a bias voltage that almost covers the full

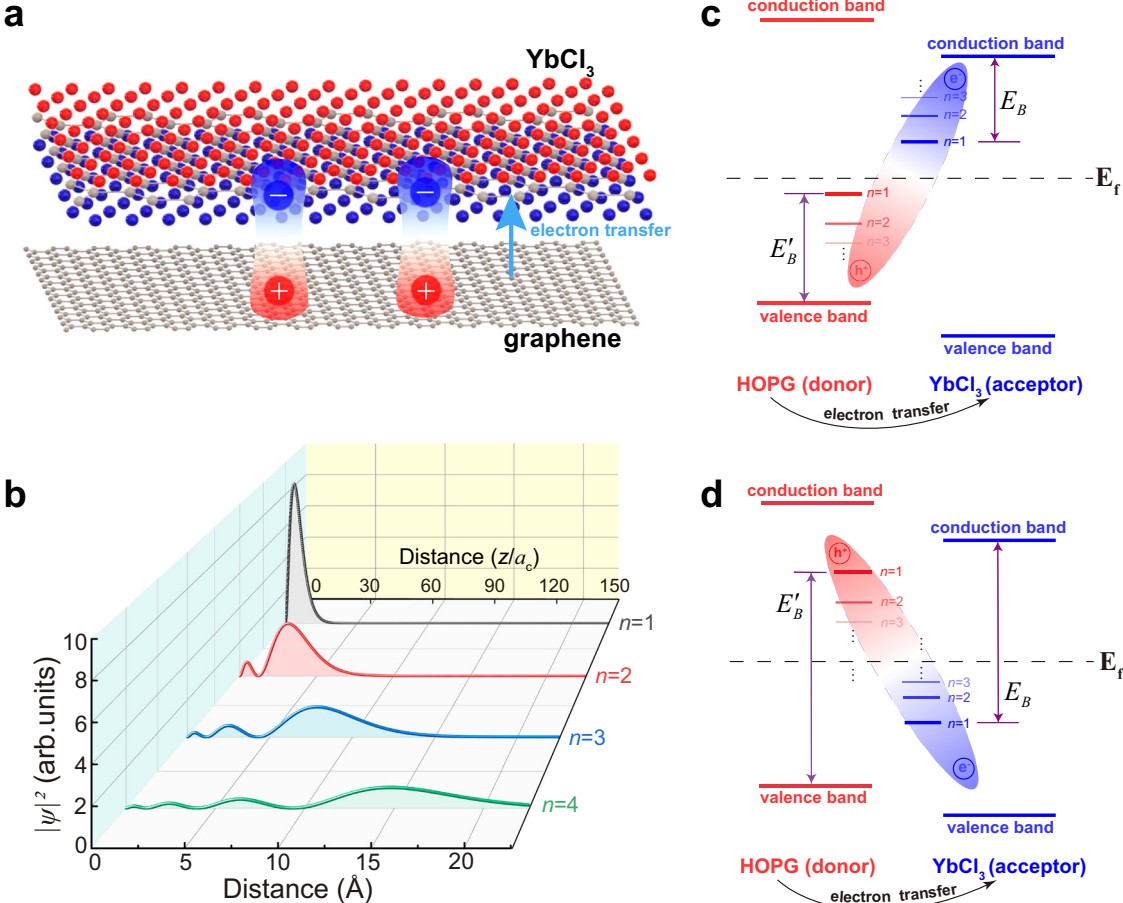

**Fig. 4 | Interlayer *e-h* bound state. a** Pictorial depiction illustrating the electron transfer from the underlying graphene to the upper YbCl₃ layer. **b** Approximate spatial distribution of probability density for one-dimensional Rydberg states of $n=1,2,3,4$ along the out-of-plane $z$ direction. The characteristic Bohr radius $a_c$ in our analysis is chosen to be 0.15 Å. **c, d** Schematic plots describing the energy-level configuration of the charge-transfer exciton with small or large binding energy, where the Rydberg series of transferred $e$ and $h$ are represented by a column of blue and red bars respectively. The energy offset between Rydberg levels and band edge scales with the binding energy (indicated by purple double arrows).

positive range of YbCl₃ gap and involves multiple Rydberg states of different $n$ into the tunneling. Therefore, contrasting to the monotonic decaying current-height curve for the normal bulk band states (see Supplementary Figs. 5 and 7), the declining and oscillating feature indicates that the experimental current-height curve is the superposition of several $I^n_{d_{3z^2-r^2}}$ or $I^n_{p_z}$, proving the Rydberg exciton nature of the in-gap states. However, seeking a precise quantitative fitting is not practical here due to the lack of microscopic information of the tunneling junction, such as the exact apex shape and electronic structure of the tip and Rydberg series.

Figure 4c, d displays two energy-level diagrams of charge-transfer exciton resulting from different sizes of binding energy. When altering the size and polarity of bias voltage the tunneling-active states change accordingly, fueling the opportunity of inspecting the shape evolution of Rydberg charge clouds as a function of energy to discriminate between two cases. Therefore, we carry out the bias-dependent current-height characteristic measurement. Controlling the tip to repeat the same path of retraction at different bias voltages, the bias-dependent current-height curves are obtained as shown in Fig. 5d, e (see Methods, and more data obtained from YbCl₃ or other samples are shown in Supplementary Fig. 7). The tunneling current is contributed by $I^n_{d_{3z^2-r^2}}(z)$ or $I^n_{p_z}(z)$ summed over Rydberg orbital of $n$ lying in the energy window $E_U$. Changing the bias alters the width of $E_U$ and corresponding tunneling-active Rydberg $n$-states, leading to different landscapes of the current–height curve. Hence the shape

evolution of Rydberg charge clouds as a function of energy can be resolved. As shown in Fig. 5d, the current-height curve obtained at 0.8 V exhibits a composite shape that encompasses both the smooth, nodeless structure of the small $n$ state and the pronounced undulations associated with the large $n$ state. As the bias is reduced to near the Fermi level, reaching 0.1 V, the $E_U$ is significantly narrowed, and the curve undergoes a gradual transformation into a shape characterized by prominent undulations with two nodes, indicating a principal quantum number of at least three. This pattern signifies the predominant contribution of the large $n$ state located in the vicinity of the Fermi level. Thus, it is found in the positive bias range that smaller $n$ states occupy higher energy positions, while larger $n$ states occupy lower energy positions. This result demonstrates that it is the $h$-Rydberg levels lying above the Fermi level that are probed at positive bias, as their principal quantum number $n$ descends with increasing energy. However, there is an untouchable space above the interface since the tip cannot reach too intimate contact with the sample. Thus, whether Rydberg series with $n$ larger than 3 are occupied and detected or not cannot be surely inferred from the shape of $I$–$z$ curves. Vice versa, the ascending $n$ of $e$-sequence is also observed below the Fermi level (Fig. 5e). Thus, the configuration of large binding energy described in Fig. 4d is confirmed to be our case. Given the insulating gap of YbCl₃ ranging from +0.8 eV to −0.8 eV, the interlayer exciton binding energy here is deemed to be unusually large up to electron-volt scale.

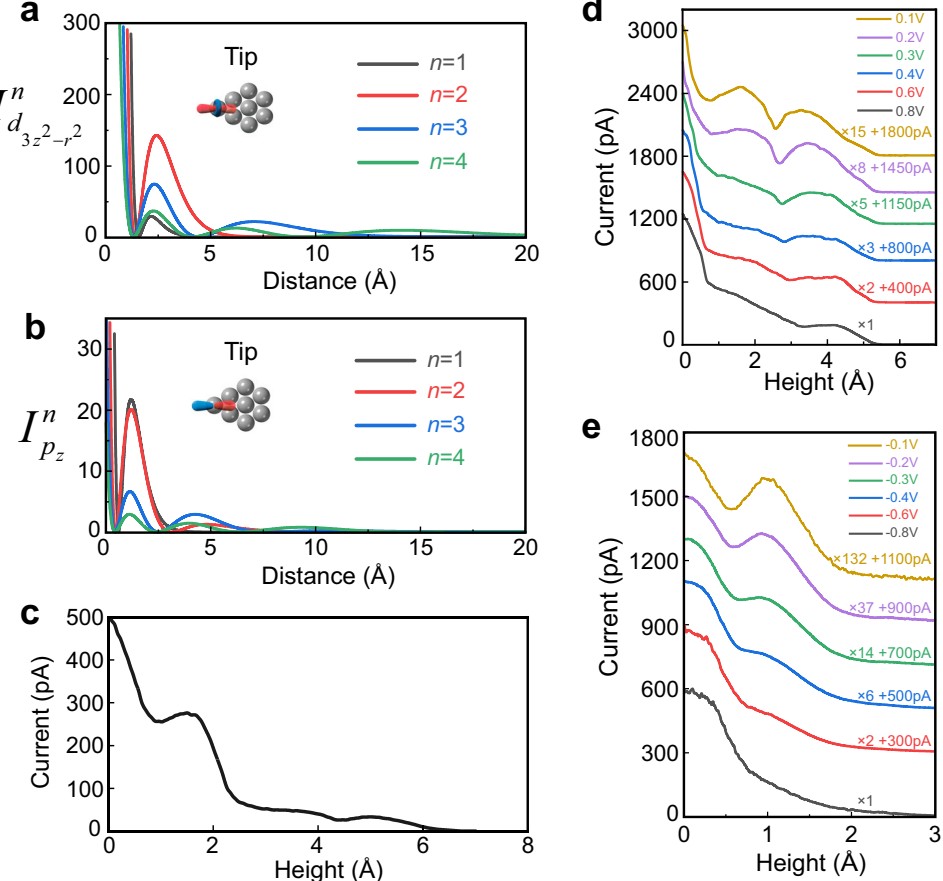

**Fig. 5 | *I–z* characteristics contributed by Rydberg series. a, b** Simulated *I–z* curves contributed by Rydberg state of *n* under the tip $d_{3z^2-r^2}$ and $p_z$ orbital, with the current axis in arbitrary unit. The inset in (**a**) and (**b**) illustrates the tip apex and corresponding $d_{3z^2-r^2}$ and $p_z$ orbital. **c** Experimental current-height curve measured at 0.7 V. The zero-height point is determined by the setup condition of $U = 0.7$ V and $I = 500$ pA, which is different from the zero-distance point in (**a**) and (**b**) simulations. **d, e** Bias-dependent current-height characteristics. By switching off the STM feedback system and artificially controlling the tip to repeat the same path of retraction several times under different bias voltages, the current as a function of tip height under a certain bias is obtained. To clearly illustrate the shape, the curves are magnified and offset with the corresponding factor indicated for each curve.

Noted that the size of binding energy is in line with the extent of localization of *e/h* wavefunction[41], the large interlayer binding energy can be ascribed to the unusually *e-h* tight binding that roots from the highly localized Yb-4f character. Moreover, importantly, considering the tunneling principle (see Methods), the large binding energy which renders an energy-level configuration as shown in Fig. 4d is indispensable to make the charges of exciton complexes detectable via local tunneling probe.

In conclusion, we directly show, for the first time, the atomic-scale visualization and characterization of the charge-transfer exciton complexes at the interface between YbCl₃/graphene via STM. Our results elaborate the electronic structure of correlated insulating YbCl₃ monolayer and in-gap excitonic states, and demonstrate the capability of STM in profiling the spatial distribution of Rydberg charge density. The *e-h* bound states across the interface reside in both the YbCl₃ monolayer and graphene underlayer, they thus suffer the strong impact of interlayer moiré periodic potential, which manifests in our experiment as the moiré-determined periodic orders of exciton complexes [Fig. 2 and Supplementary Fig. 3] that could be an indication of moiré-trapped excitons[14,17,27]. In addition, the exciton complexes experience strong mutual Coulomb correlation, contributed by both the poor screening of dilute transferred charges and the highly correlated nature of 4f orbital. The interplay and rivalry of various degrees of freedom may foster unique many-body orders, while our system with substantial binding energy offers a robust ground-state exciton ensemble to explore at the atomic scale.

## Methods

### MBE growth
The YbCl₃ monolayer was grown on cleaved HOPG substrate by MBE (molecular beam epitaxy) at a base pressure of ~ $1 \times 10^{-9}$ mbar. The commercial anhydrous YbCl₃ powder was evaporated at 680 °C and the substrate was kept at 340 °C during the growth. The process of growth was monitored by reflective high-energy electron diffraction. The YbCl₃ monolayer coverage reaches roughly 60% after 30 minutes of deposition.

### STM measurements
Experiments were performed with a home-built hybrid STM and MBE system, and the STM measurements were operated at a temperature of 5 K. The sample was directly transferred into STM to perform in situ measurement after finishing the growth. Scanning tunneling spectroscopy was performed using a lock-in amplifier technique with a modulation frequency of 963 Hz and a root-mean-square modulation voltage between 5 mV and 20 mV depending on the spectral range of interest.

### DFT computations
The DFT calculations for YbCl₃ monolayer were carried out to elucidate the electronic structure and make a comparison with experimental dI/dV spectra, using the full-potential augmented plane wave plus local orbital code (Wien2k)[42]. According to the experimentally resolved lattice constant of $6.65 \pm 0.1$ Å, we

adopted the YbCl₃ lattice parameters of $a = b = 6.65$ Å for computation and optimized the atomic positions. The plane wave cutoff energy of 14 Ry was set for the interstitial wave functions, and 11*11*1 k-mesh was used for integration over the Brillouin zone. The muffin-tin sphere radii were chosen to be 2.5 and 2.0 Bohr for Yb and Cl atoms, respectively. We performed LSDA + U + SOC calculations with the typical values Hubbard $U = 8.5$ eV and Hund exchange $J_H = 1.0$ eV for Yb 4 f electrons[43], and the spin–orbit–coupling (SOC) effect was included by the second-variational method with scalar relativistic wave functions.

## Current-height characteristic

Before we take the current-height characteristic, the setup condition described in Fig. 5c–e is adopted so that the tip is positioned over the sample at a relatively close distance in order to ensure that the interfacial excitonic states can be accessed by the tip. Then disable the feedback to fix the tip at current position which acts as the zero-height start point in the curve recorded later. Next step is to retract the tip and, in the meantime, record the tunneling current so the current–height curve is obtained. The bias can be set to the chosen value during the backward process of the tip, therefore the bias-dependent current-height characteristic can be acquired.

## Work function and interfacial charge transfer

The work function of YbCl₃ monolayer is calculated to be 7.1 eV, which is larger than that of 4.6 eV for graphene, suggesting a charge transfer across the YbCl₃/graphene interface. In our computation, YbCl₃ monolayer is slightly electron doped when contacting graphene, and the electron doping level is estimated to be 0.21 e/nm² by Bader charge analysis[36]. In realistic system, the electrostatic field of the various excitonic arrays associated with different moiré potentials would alter the work function difference and the consequential amount of charge transfer accordingly.

## Analysis of one-dimensional Rydberg series

The wavefunction of interlayer charge-transfer exciton is very analogous to the well-known image potential state. Charges in front of a conducting metal surface experience an attractive Coulombic force which is caused by the opposite mirror charges induced by surficial metallic screening, leading to the bound states called IPS near the metal-vacuum interface[44–46]. Referring to the analysis from refs. 44–46 where the surficial electrostatic potential, wavefunctions and eigenvalues of IPS are comprehensively addressed, the one-dimensional Rydberg states can be analytically expressed by

$$\psi_n(z) \propto z\psi_n^{hydrogen}\left(\frac{z}{4}\right), n = 1, 2, 3, \ldots \quad (1)$$

where $z$ is the out-of-plane distance, $\psi_n^{hydrogen}(z)$ represents the well-known wavefunctions of all possible s-like ($l = 0$) bound states of hydrogen atom, and the Bohr radius $a_0$ in $\psi_n^{hydrogen}$ is replaced by the "characteristic Bohr radius" $a_c$ here. In addition, based on the numerical eigenvalues computed for a variety of electrostatic models in ref. 45, the Rydberg energy levels roughly conform to the expressions:

$$E_n^e \approx E_C^{YbCl_3} - \frac{E_B}{n^2}, n = 1, 2, 3, \ldots \quad (2)$$

$$E_n^h \approx E_V^g + \frac{E_B'}{n^2}, n = 1, 2, 3, \ldots \quad (3)$$

where $E_n^e$ and $E_n^h$ are the $n$th eigenenergy of e- and h- series, and $E_C^{YbCl_3}$ and $E_V^g$ are the minimum energy of YbCl₃ conduction band and maximum energy of graphene valence band, respectively. The reference level in Eqs. (2) and (3) is replaced from the vacuum level to the band edge noting the difference that the IPS is formed by trapping free electron at the vacuum level into bound state, while charge-transfer exciton in our system derives from trapping the Bloch e (h) of conduction (valence) band into bound states by the Coulomb attraction of h (e).

Capturing the quintessential nodal structure of Rydberg wavefunctions, Eq. (1) gives a straightforward approximation to describe the exciton states with the size of $a_c$ exclusively determining the spatial extension of charge clouds. However, the realistic circumstance is more intricate suffering various levels of complexity. (i) From the electrostatic perspective, the interfacial charges are subjected to a complicated electrostatic environment where the semimetal graphene is covered with an ultrathin YbCl₃ dielectric monolayer in proximity to the vacuum, bearing the influence of multiple parameters such as the charge numbers of charge-transfer exciton, Tomas-Fermi screening length of HOPG, dielectric constant and electron affinity of YbCl₃ monolayer. As discussed in refs. 44,45, the combination of multiple parameters dominates corresponding electrostatic constant, metallic response and dielectric response, giving rise to many sophisticated cases with disparate landscapes of effective potential and resultant wavefunction. (ii) From the orbital perspective, on the one hand, the transferred e and h populate the YbCl₃ conduction band and graphene valence band respectively, they are therefore imparted with the features of Yb-4f and C-2p orbital. This means that besides the mutual Coulomb bound potential between e and h, they are also subjected to the ionic potential and submit to the atomic orbital physics, which imposes huge influence on the real-space character of exciton states, especially noting that the Yb-4f orbitals are extremely localized and highly correlated with heavy effective mass and strong spin-orbit coupling. On the other hand, acting as extra charges embedded into an almost electroneutral crystalline background, the transferred e and h feel much mitigated Yb- and C- ionic potential because the initial valence electrons screen the ion cores, producing difficulties for assessing the exact orbital influence on wavefunction. (iii) From the many-body perspective, being different from the case of electron trapped in a constant one-dimensional potential, the bound e and h here form interactively equilibrated many-body state under the e-e, h-h, and e-h interactions. Among them, besides the pronounced e-h interaction, the e-e interaction is also particularly strong owing to the highly correlated 4 f nature, which could, in conjunction with the in-plane lattice potential and moiré potential, covert the in-plane homogenous transferred charges to lateral-organized ordering states. This is also implied by the phenomenal moiré modulation exhibited in Figs. 2 and 3 and Supplementary Fig. 3, suggesting that the transferred charges form a lattice-like lateral array with exciton complexes concentrating on each site. If so, the exciton complexes clustering around a specific site would diffuse to the vacuum hemispherically, leading to a much faster quasi-three-dimensional decaying of probability density compared to the one-dimensional model. Therefore, we steer to the experimental direction and, based on the STM-resolved z-direction charge density distribution [Fig. 5c], choose $a_c$ to be 0.15 Å which makes the charge clouds diffuse towards the vacuum to a few angstroms.

## Analysis of current-distant curves contributed by Rydberg series under specific tip orbitals

STM has the most prominent advantage of atomic-scale lateral resolution, because the actual tunneling active state of the tungsten tip is the $d_{3z^2-r^2} - p_z$ hybridized dangling bond instead of the extending state which largely diffuses to the full metallic tip as assumed in Tersoff-Hamann model[35]. The full width at half maximum is narrowed

down to 1–2 Å[39,47], yielding an extreme lateral and vertical resolution. Both theoretical and experimental studies also demonstrate the capability of STM to profile the subtle orbital configuration and texture by scrutinizing the tip-height dependence[48,49]. Taking advantage of this, the Rydberg charge densities are profiled utilizing the measurements of the current-height characteristics. However, it's needed to be pointed out that while the tip s-orbital faithfully reflects the sample wavefunction, the tip p- or d-orbital would mispresent the sample wavefunction based on the derivative rule of tunneling matrix[50]. When positioning the tip at a small tip-sample separation and fixing the bias to an in-gap value to probe the Rydberg excitons, the tunneling current as a function of $z$ as withdrawing the tip away from the interface is obtained. Considering the leading $d_{3z^2-r^2}$ and $p_z$ terms for a realistic tip, the resultant current-distance characteristic can be expressed by Chen's rule[50,51]:

$$I_{d_{3z^2-r^2}}(V,z) = \sum_{E_n \in E_U} I^n_{d_{3z^2-r^2}}(z) \propto \sum_{E_n \in E_U} C_n \left| M^n_{d_{3z^2-r^2}}(z) \right|^2$$
$$\propto \sum_{E_n \in E_U} C_n \left| \left( 3\kappa^{-2} \frac{\partial^2}{\partial z^2} - 1 \right) \psi_n(z) \right|^2 \quad (4)$$

$$I_{p_z}(V,z) = \sum_{E_n \in E_U} I^n_{p_z}(z) \propto \sum_{E_n \in E_U} D_n \left| M^n_{p_z}(z) \right|^2 \propto \sum_{E_n \in E_U} D_n \left| \frac{\partial}{\partial z} \psi_n(z) \right|^2 \quad (5)$$

where the setup bias voltage $V$ spans the energy-integrating window $E_U$, inside which the Rydberg state of $n$ fulfilling $E_n \in E_U$ contributes to tunneling current $I^n_{d_{3z^2-r^2}}(z)$. $I^n_{d_{3z^2-r^2}/p_z}(z)$ and $M^n_{d_{3z^2-r^2}/p_z}(z)$ are the tunneling current and matrix of $n$ state under the $d_{3z^2-r^2}$ and $p_z$ tip orbital, respectively. The coefficients $C_n$ and $D_n$ depend on subtle tunneling conditions, and the decay constant $\kappa$ is determined by the work function $\varphi$ through the relation $\kappa = (2m_e\varphi)^{\frac{1}{2}}/\hbar \approx 0.51\sqrt{\varphi}$, in Å and eV. For simplicities, the other-than-$d_{3z^2-r^2}/p_z$ terms of the tip and the hybridization between $d_{3z^2-r^2}$ and $p_z$ orbitals are neglected.

Using the regular $\varphi$ of 4 eV and Eqs. (4) and (5), the respective current contributions $I^n_{d_{3z^2-r^2}}$ and $I^n_{p_z}$ as a function of $z$ are plotted in Fig. 5a, b.

### Tunneling accessibility towards charges of exciton complex

We report the interfacial charge-transfer excitons which appear as the in-gap states in tunneling spectrum featuring the Rydberg-type spatial wavefunction. Since charge transfer is a rather common phenomenon which occurs for almost all heterostructures, one would expect such interfacial in-gap states to be universal in STM experiments. However, the relevant report is very seldom, leading to the question why many STM studies with regard to low-dimensional heterostructure do not show excitonic in-gap states. In terms of the principle of tunneling we address this issue as follows.

For convenience of the following discussion, we assume the excitonic state is the bound pair comprised of one $e$ and one $h$. In most material heterostructures the interlayer binding energy ranges from tens to hundreds of millielectron volts[30,52], thus, the energy-level configuration conforms to the diagram of small binding energy as shown in Fig. 4c where the transferred $e$-series lie above the Fermi level with the $n = 1$ state occupied by a transferred electron in the ground state (similarly, the $h$-states lie below the Fermi level with $n = 1$ state occupied by a transferred hole). At positive sample bias, the electrons tunnel from tip to sample with the Rydberg $e$-series included in $E_U$. However, in this case the $e$-series can not serve as the tunneling destination because they are excitation levels for the transferred electron rather than the sample empty state prepared for accepting tunneling electrons. To be more specific, the injection of another $e$ from the tip

alters the $1e$–$1h$ bound state into the renormalized $2e$–$1h$ bound state, namely the trion state[7], which would be excited at the bias threshold slightly lower than the conduction band minimum by a small trion binding energy of about tens of millielectron volts[3]. Vice versa, all the transferred $h$-series below the Fermi level cannot be sensed directly, but only in an indirect manner of the $1e$–$2h$ trion formation when injecting the tunneling $h$ at negative bias voltage.

When it comes to the large binding energy case illustrated in Fig. 4d, things turn different as the energy configuration renders the charge transfer states accessible in tunneling process. Since the $h$-series are situated above the Fermi level, they are substantially occupied by transferred holes and can thus accept the tunneling electrons at positive sample bias, and vice versa for the $e$-series below the Fermi level as they are occupied by transferred electrons. At positive bias with electrons flowing from tip to sample, the tunneling electron annihilates the transferred $h$ and destroys an interlayer $e$-$h$ pair. Since the interlayer $e$-$h$ pairs are energetically favored ground state, the tunneling electron which annihilates the $e$-$h$ pair will leave the initial $h$ state for underlying graphene rapidly owing to the fast interlayer charge transfer. The ultrafast interlayer charge transfer would take place within a time length of ~5 fs[53], which is remarkably much shorter than the nanosecond-scale flowing time of tunneling electron under the tunneling current set up to hundreds of picoamperes, eventually restoring the interlayer $e$-$h$ pair and giving rise to steady tunneling current. Consequently, the charge-transfer excitons directly participate in the tunneling process and produce the in-gap conductance.

In the measurement shown in Fig. 5d, e, at negative bias the electron tunnels out from the occupied exciton states of the sample to tip, displaying an ascending $n$ sequence of $e$-series. When injecting tunneling electrons into the empty states of the sample at positive bias, the $n$ sequence switches to the descending $h$-series immediately. Such a sudden change with respect to the bias polarity is consistent with the tunneling regime where switching the bias polarity immediately reverses the direction of electron tunneling. This result also aligns with the judgment that all the Rydberg $e$-states lying below the Fermi level are occupied (so do the $h$-states above the Fermi level), ensuring that there are indeed electrons that could tunnel out from the occupied Rydberg $e$-series at negative bias, and that could tunnel into the empty $h$-series at positive bias. A corollary is that several Rydberg states of different $n$ are occupied by $e/h$, resulting in the bound state of multiple electrons and holes, i.e., exciton complex, whose wavefunctions resemble, let's say, Helium orbitals, Lithium orbitals and so on.

## Data availability

The data generated during this study are available within the article and the Supplementary Information file. Source data are provided with this paper.

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

## Acknowledgements

We thank Wulf Wulfhekel and Xiaopeng Li for their insightful discussions. C.G. acknowledges funding from the National Key Research and Development Program of China (Grant nos. 2019YFA0308404, 2022YFA1403301), National Natural Science Foundation of China (Grant nos. 11427902,12274084, 12241402), Innovation Program of Shanghai Municipal Education Commission (Grant no. 2023ZKZD03), Science and Technology Commission of Shanghai Municipality (Grant nos. 20JC1415900, 23JC1401100) and Shanghai Municipal Science and Technology Major Project (Grant no. 2019SHZDZX01). H.W. acknowledges support from the National Natural Science Foundation of China (Grants nos. 12174062 and 12241402).

## Author contributions

M.Z. fabricated the samples. M.Z., Z.W., C.W., and C.L. performed the measurements. L.L. conducted the DFT computation. Z.W., M.Z., C.W., F.Y., and C.G. analyzed the data. L.L. and H.W. conducted the

computational analysis. Z.W. and C.G. conceived the project and wrote the paper with inputs from all co-authors.

## Competing interests

All authors declare no competing interests.
