## [Peer Review File · Nature Communications]

In the manuscript by Zhao *et al.*, they reported a STM study in YbCl₃/HOPG, where they observed out-of-plane dispersion of electronic clouds and attributed the observation to the formation of Rydberg states. They suggested interlayer excitons spontaneously became the ground state through charge transfer. However, I don't find the evidence provided by the authors can exclusively lead to their conclusions. I cannot suggest publication of this work. The reasons are as follows.

(1) This manuscript only provides "hand-waving" arguments about their observations. YbCl₃ is not a familiar material to many. The authors didn't provide sufficient information about this material. In some literature, it was reported to have a fairly large bulk band gap of ~4.4 eV. I doubt in this case, any efficient charge transfer can happen between YbCl₃ and HOPG. Are there any other signatures of interlayer charge transfer other than inferring from calculated workfunction and bandgaps?

(2) For any bound state without any excitation, it should stay in the ground state at the base temperature. I don't think the observation of distance dependent dI/dV is due to the excited Rydberg series.

(3) The authors seem to give up analyzing the features in tunneling spectroscopy but rather provide vague discussion "Changing the tip-interface distance is supposed to alter the relative ratio of Rydberg components that participate in the tunneling process, leading to a distinct dI/dV spectrum". So based on their argument, the detailed dI/dV spectrum is nothing important and provide no further information. In my point of view, the dI/dV spectra have a lot of details and should be looked into more carefully.

(4) Overall, I doubt the Rydberg states is the only explanation of the experimental observation. The in-gap states could be due to the moiré effects on the electronic dispersion of HOPG. It's well known that the van Hove singularities can emerge in the moiré superlattices of graphene/hBN and twisted bilayer graphene. The same physics can happen here and gives rise the peaks in dI/dV. As the moiré with different periodicity can have van Hove singularities at different energies, it certainly can cause a height difference with certain bias and current across the interface between two domains. The oscillatory feature with retracting the tip could be results of spatial extension of different moiré bands.

REVIEWER COMMENTS

Reviewer #2 (Remarks to the Author):

Zhao et al. discuss atomic scale visualization of Rydberg excitonic complexes formed at the interface of YbCl₃/HOPG. I find this an interesting interface to investigate as YbCl₃ is a strongly correlated system. They use STM to elucidate the Rydberg physics at the interface. As such they observe changes in the imaging as well as the dI/dV spectra which is followed by complex modelling to understand the physics.

I find the work to be compelling but needs major revisions to meet the standards for publication in Nature Communications.

Firstly, the figures need to be improved or broken into 4 or 5 figures. They are a bit clumsy now and combine too many messages in the same panel. Fonts need to be bigger and readable.

Secondly, while the overall message is to some degree clear, I find the modelling a bit too much involved compared to the data. More processing on the data or some free-parameter fits to the dI/dV plots would make more sense rather than fixed parameter simulations that qualitatively match the data. This, in my view, is crucial as I do not feel the current structure of the paper is strong enough to support the claims.

To elaborate, they do not visualize any "excitonic complex" in imaging mode but is inferred from dI/dV plots. It is also not clear the full Rydberg series (up to n=4) is seen in the data (Fig. 3).

Finally, I recommend the authors to add some more high resolution scans of the Moire' lattice at different biases to help the readers understand better.

Some other minor comments :

1. The choice of the word "negative band gap" is unnecessary in the abstract and a bit misleading.

2. Could there be broadening of the V-shaped dI/dV curve due to imperfect interfaces leading to a parabolic shape of the same?

3. Could the authors elaborate on the tip-apex change a bit more quantitatively?

4. Perhaps the authors can also expand on what they mean by "slits" in the sample? Is it introduced during the growth?

We thank the referees for their assessment and valuable comments/suggestions which help us a lot to improve our manuscript. Here, we answer point by point to the concerns or suggestions of the referees.

Response to the reviewer #1

In the manuscript by Zhao *et al.*, they reported a STM study in YbCl₃/HOPG, where they observed out-of-plane dispersion of electronic clouds and attributed the observation to the formation of Rydberg states. They suggested interlayer excitons spontaneous became the ground state through charge transfer. However, I don't find the evidence provided by the authors can exclusively lead to their conclusions. I cannot suggest publication of this work. The reasons are as follows.

Reply #1A:

We sincerely thank the reviewer for the careful reading and assessment of our manuscript. The reviewer's summary is accurate. The key point of this work is to unveil the charge transfer exciton by observing the large and oscillating electronic clouds, especially considering that such a large size of charge clouds corresponds to atomic orbitals with very low nuclear charge number Z , such as hydrogenic or helium orbitals. Which is described in the main text (page 9, line 10 in the revised manuscript):

...Considering that the mean radius of atomic orbital is inversely proportional to the nuclear charge number Z , the large wavefunction extent represents hydrogenic-like orbitals with very small Z , which is inconsistent with the tremendous Z of Yb and Cl ions.

The reviewer raises important concerns, of which the related physics regarding charge-transfer process, Rydberg orbitals and moiré singularities shall be elaborated. In our humble opinion, the concerns can be safely lifted after careful reexamination. Below, we provide detailed discussion towards the physical mechanism and the reviewer's concerns.

This manuscript only provides “hand-waving” arguments about their observations. YbCl₃ is not a familiar material to many. The authors didn't provide sufficient information about this material. In some literature, it was reported to have a fairly large bulk band gap of ~4.4 eV.

Reply #1B:

YbCl₃ is a relatively rarely explored material and thus unfamiliar to many researchers. Of particular importance is the rare-earth 4f¹³ orbital occupation of Yb³⁺ ions and the charge transfer insulator nature. The main information we provide based on DFT computation is presented in Extended Data Fig.1. The huge Hubbard U of highly localized 4f electrons splits a tremendous Mott gap, and the lower Hubbard bands lie greatly lower than the first valence band derived from Cl-3p orbitals, rendering the material a charge transfer insulator.

As the reviewer mentioned, we learned that there is an optical measurement showing a ~4.4 eV gap for bulk YbCl₃, while our STS measurement reveals a full gap of ~1.6 eV for monolayer YbCl₃ deposited on HOPG substrate. We think this is a common phenomenon since the gap size of correlated insulator YbCl₃ is highly determined by the Hubbard U , while the metallic HOPG substrate provides an electrostatic environment with high electronic screening that could

reduce the Hubbard U significantly. There are many examples of band gap reduction for monolayer insulator on metallic substrate, even on the magnitude of electron volts [1-3].

...I doubt in this case, any efficient charge transfer can happen between YbCl_3 and HOPG.

Reply #1C:

If we didn't get the reviewer wrong, this is one of the most common misunderstandings. A lot of researchers hold the physical picture that the HOPG electrons at the Fermi level jump to the conduction band of YbCl_3 by overcoming the band gap of the latter. Considering the large gap barrier, the chances for charge transfer is near zero without significant thermal agitation. The larger the band gap is, the lower the chances are.

Above misunderstanding stems from the ignorance of the fact that the alignment of two Fermi levels (chemical potential for a gapped material) is itself a result of interfacial charge transfer [4]. As shown in Fig. R1a, before the contact, graphite and YbCl_3 share the same vacuum level E_0 , with the work function of 4.6 eV and 7.1 eV respectively. If not contacted, such a huge work function difference makes the graphite chemical potential higher than that of YbCl_3 . When contacting these two materials closely, the work function difference induces a great built-in electric field on the magnitude of $\text{eV}/\text{\AA}$, which drives substantial electrons to transfer (Fig. R1b). The electrons (holes) transferred to YbCl_3 (graphite) elevate (lower) the band energy position, eventually leading to a balance. As seen in Fig. R1c, after the contact, due to that the electric field induced by work function difference has already been compensated by the opposite electric field of interlayer transferred charge, the two materials finally reach the same chemical potential (Fermi level).

Fig. R1 Schematic diagrams showing the charge transfer process.

Thus, as long as the work function difference exists, the interfacial charge transfer is inevitable when closely contacting them, no matter how large the band gap is. For our case with such a significant work function difference, considerable charge transfer is reasonable and indispensable, which is consistent with our DFT results and many other works [5,6].

...Are there any other signatures of interlayer charge transfer other than inferring from calculated workfunction and bandgaps?

Reply #1D:

We understand that at current stage the reviewer doubts the Rydberg orbitals and considers moiré states distribution and other possibilities as candidate explanations. However, the very large spatial extent (several angstroms to nanometer scale) and the oscillatory feature in I-z

curve is the strongest signature of small- Z Rydberg orbitals. **All alternative explanations derived from conventional band states in this system are incapable of explaining the non-monotonic oscillation and the magnitude of orbital size, owing to the highly localized Wannier orbitals of C-2p, Cl-3p and Yb-4f characters around the Fermi level.**

Being renormalized by intersite hopping, interlayer coupling or moiré potential, the band states all derive from atomic orbitals and hold the Wannier orbital characters. Through various approximation methods, it's widely acknowledged the expected orbital radius r of particular quantum number $nlm=100$ satisfies the relation $\langle r \rangle = \frac{a}{Z_{eff}}$ [7], where a is the hydrogenic Bohr radius of 0.53 Å and Z_{eff} is the effective nuclear charge number, which is proportional to Z but reduced to some level due to the screening of other electrons [7]. Therefore, the orbital radius diminishes rapidly as the nuclear charge number Z becomes larger. With the smallest Z and biggest $\langle r \rangle$, the largest hydrogenic orbitals of $n=3, 4$ extend to 10-20 a in space, i.e., 5-10 Å. Compared to the hydrogenic case, **orbitals of elements with higher Z are significantly localized, completing the large- n nodal oscillation in a very limited region and leaving only the exponentially decaying tails accessible to the STM measurement.** So in most STM I-z experiments carried on crystal surface, as a rule of thumb, the tunneling current decays monotonically by one order of magnitude per 1 Å [8].

The extended charge clouds with non-monotonic undulation signify a very mitigated out-of-plane trapping potential related to small Z , giving rise to small energy level spacing and large spatial distribution, for which none of the Wannier orbitals of C-2p, Cl-3p, Yb-4f can fit.

For any bound state without any excitation, it should stay in the ground state at the base temperature. I don't think the observation of distance dependent dI/dV is due to the excited Rydberg series.

Reply #1E:

The reviewer is right that any bound state should stay in the ground state without excitation, and this is also our claim in the manuscript. Instead of being caused by thermal or other types of excitation, **the distance-dependent observations are results of the Rydberg ground state due to the charge transfer and significant binding energy.** It's kind of an analogy to the direct exciton in semiconductors, where the exciton transitions from excited state to spontaneous ground state when the binding energy exceeds the band gap [9-11]. As depicted in the energy level diagram of Fig. 3d in the original version (Fig. 4d in the revised version), some electron levels of $n=1, 2 \dots$ lie below the Fermi level, so they should be occupied at the ground state, and so do the hole levels above the Fermi level. The conclusion is that **the ground state is the Rydberg series.** Relevant detailed discussion has been provided in the *Tunneling accessibility towards charges of exciton complex* paragraph in the **Methods** section.

The authors seem to give up analyzing the features in tunneling spectroscopy but rather provide vague discussion "Changing the tip-interface distance is supposed to alter the relative ratio of Rydberg components that participate in the tunneling process, leading to a distinct dI/dV spectrum". So based on their argument, the detailed dI/dV spectrum is nothing important and provide no further information. In my point of view, the dI/dV spectra have a lot of details and should be looked into more carefully.

Reply #1F:

We agree with the reviewer that the dI/dV spectrum contains much information, but we are dealing with completely different target states with unprecedentedly large orbital sizes and distinct spatial occupations of different n -states (Fig. 4b in the revised version or Fig. 3b in the original version). Unlike conventional surficial states detected in STM where only the exponentially decaying tails contribute to tunneling current and the dI/dV features are to a great degree insensitive to the tip-sample distance, the tunneling matrix here is highly changeable and unpredictable as a function of tip-sample distance, due to the high dispersion of Rydberg n states and the absence of the full understanding of the tip orbital and band structure. Suffering the multiple levels of complexes listed in Reply #2C to Reviewer #2, to decipher the complicated information of the dI/dV spectrum is a challenging task beyond the scope of this manuscript, while this study focuses on elucidating the existence of interfacial Rydberg states.

Overall, I doubt the Rydberg states is the only explanation of the experimental observation. The in-gap states could be due to the moiré effects on the electronic dispersion of HOPG. It's well known that the van Hove singularities can emerge in the moiré superlattices of graphene/hBN and twisted bilayer graphene. The same physics can happen here and gives rise the peaks in dI/dV. As the moiré with different periodicity can have van Hove singularities at different energies, it certainly can cause a height difference with certain bias and current across the interface between two domains. The oscillatory feature with retracting the tip could be results of spatial extension of different moiré bands.

Reply #1G:

The reviewer is right that moiré singularities lead to distinct band dispersion and LDOS structure, and it can be one of the explanations for the height difference across domains under a certain imaging bias. Nevertheless, it fails to explain the oscillation and large charge cloud presented in the I-z curve. The reasons are as follows.

- (i). Determined by the periodically varying local atomic stacking structure, the moiré pattern serves as an **in-plane (x-y plane)** periodic potential, which modulates the **in-plane** wavefunction distribution [12-15], and is inconsistent with our observation of spatial dispersion along **z-direction**. The van Hove singularities opened at the moiré Brillouin zone (also a two-dimensional in-plane Brillouin zone) occupy different in-plane spaces with respect to the moiré potential landscape, which has been experimentally confirmed [13,16,17]. Besides, the charge density simulation in a moiré heterostructure also doesn't exhibit out-of-plane modulation for both layers [18].
- (ii). As mentioned in above discussion concerning orbital radius, the graphene moiré singularities derive from C-2p_z orbital renormalized by moiré structure, whose charge clouds are inherently localized [18,19] and cannot support the large spatial extension at z-direction.

References

1. K. T. Winther and K. S. Thygesen. Band structure engineering in van der Waals heterostructures via dielectric screening: the G Δ W method. *2D Materials* **4** (2017).
2. L. Waldecker *et al.* Rigid Band Shifts in Two-Dimensional Semiconductors through

- External Dielectric Screening. *Phys. Rev. Lett.* **123**, 206403 (2019).
3. S. Ulstrup *et al.* Ultrafast Band Structure Control of a Two-Dimensional Heterostructure. *ACS Nano* **10**, 6315-6322 (2016).
 4. M. Yoshitake. Work Function and Band Alignment of Electrode Materials. *NIMS Monographs* (2021).
 5. L. Xu, T. Wu, P. R. C. Kent, and D.-e. Jiang. Interfacial charge transfer and interaction in the MXene/TiO₂ heterostructures. *Physical Review Materials* **5**, 054007 (2021).
 6. S. Biswas *et al.* Electronic Properties of α -RuCl₃ in Proximity to Graphene. *Phys. Rev. Lett.* **123**, 237201 (2019).
 7. D. J. Griffiths and D. F. Schroeter, *Introduction to quantum mechanics* (Cambridge university press, 2018).
 8. S. Lounis. Theory of scanning tunneling microscopy. *arXiv preprint arXiv:1404.0961* (2014).
 9. A. Kogar *et al.* Signatures of exciton condensation in a transition metal dichalcogenide. *Science* **358**, 1314-1317 (2017).
 10. W. Kohn. Excitonic Phases. *Phys. Rev. Lett.* **19**, 439-442 (1967).
 11. Y. Jia *et al.* Evidence for a monolayer excitonic insulator. *Nature Physics* **18**, 87-93 (2021).
 12. H. Yu *et al.* Moiré excitons: From programmable quantum emitter arrays to spin-orbit-coupled artificial lattices. *Science advances* **3**, e1701696 (2017).
 13. M. Yankowitz *et al.* Emergence of superlattice Dirac points in graphene on hexagonal boron nitride. *Nature Physics* **8**, 382-386 (2012).
 14. C. Zhang *et al.* Interlayer couplings, Moiré patterns, and 2D electronic superlattices in MoS₂/WSe₂ hetero-bilayers. *Science advances* **3**, e1601459 (2017).
 15. S. Carr, S. Fang, and E. Kaxiras. Electronic-structure methods for twisted moiré layers. *Nature Reviews Materials* **5**, 748-763 (2020).
 16. G. Li *et al.* Observation of Van Hove singularities in twisted graphene layers. *Nature Physics* **6**, 109-113 (2010).
 17. Z. Li *et al.* Observation of van Hove Singularities in Twisted Silicene Multilayers. *ACS Cent. Sci.* **2**, 517-521 (2016).
 18. J. Kang *et al.* Electronic structural Moire pattern effects on MoS₂/MoSe₂ 2D heterostructures. *Nano Letters* **13**, 5485-5490 (2013).
 19. M. H. Naik and M. Jain. Substrate screening effects on the quasiparticle band gap and defect charge transition levels in MoS₂. *Physical Review Materials* **2** (2018).

Response to the reviewer #2

Zhao et al. discuss atomic scale visualization of Rydberg excitonic complexes formed at the interface of YbCl₃/HOPG. I find this an interesting interface to investigate as YbCl₃ is a strongly correlated system. They use STM to elucidate the Rydberg physics at the interface. As such they observe changes in the imaging as well as the dI/dV spectra which is followed by complex modelling to understand the physics.

I find the work to be compelling but needs major revisions to meet the standards for publication in Nature Communications.

Reply #2A:

We are very grateful to the reviewer for the assessment and vital suggestions. We have revised the manuscript or provided further discussion according to the reviewer's suggestions. We hope these revisions and information can enhance the strength of this study.

Firstly, the figures need to be improved or broken into 4 or 5 figures. They are a bit clumsy now and combine too many messages in the same panel. Fonts need to be bigger and readable.

Reply #2B:

We agree with the reviewer and appreciate the helpful suggestion. Following the advice, both Fig. 2 and Fig. 3 have been split into two figures, respectively, and corresponding fonts and labels are improved in the revised version. Now there are 5 figures in total, and associated paragraphs have been reorganized to match the new version. The modifications in paragraph configuration are labeled by blue fonts in the parentheses in the revised manuscript.

Secondly, while the overall message is to some degree clear, I find the modelling a bit too much involved compared to the data. More processing on the data or some free-parameter fits to the dI/dV plots would make more sense rather than fixed parameter simulations that qualitatively match the data. This, in my view, is crucial as I do not feel the current structure of the paper is strong enough to support the claims.

Reply #2C:

We agree with the reviewer that there is plenty of valuable quantitative information about Rydberg physics that can be excavated beyond qualitative comparison. We try our best to obtain a multiparameter fitting using the model described later. However, a lot of approximations have to be invoked in quantitative analysis due to the lack of full understanding of the tunneling junction, especially the microscopic nature of tip apex and the tip-sample interaction, which is the common challenge confronted by all the STM measurements. To make a brief summing-up, as can be seen later, the fitting results are still qualitatively plausible, yet incapable of unraveling the underlying intricate complexes quantitatively. As explained later, the multiple levels of complexities involve rich microscopic and many-body information. Some of them are hard to access while others require systematic theoretical and experimental investigation. Despite the intriguing physics beneath it, we regret to say it's too sophisticated for us at the current stage. To avoid any kind of misleading, we did not include this fitting into our manuscript.

The fitting model, results and the discussion of the intertwined difficulties are listed below:

We approximate the tip states as hybridized d_{zz} and p_z orbital with quenched orbital momentum and neglect the contributions of other in-plane components. The tip state is:

$$|\psi_t\rangle = D|d_{zz}\rangle + P|p_z\rangle \quad (1)$$

D and P are the coefficients of corresponding d_{zz} and p_z components, respectively. Here, we set D and P as free parameters and perform the normalization at the last step when the fitting spectra are obtained. The sample wavefunction as a function of z (tip height) is treated as the superposition of Rydberg states of different n :

$$|\psi_{sam}(z)\rangle = \sum_n C_n |\psi_n(z, d, a_c)\rangle, \quad n = 1, 2, 3 \dots \quad (2)$$

C_n is the coefficient of Rydberg n -state, and d is the distance between the tip initial position (zero point in I-z spectra) and the interface. The wavefunction of Rydberg n -state is:

$$\psi_n(z) \propto z \psi_n^{hydrogen}\left(\frac{z+d}{4}\right), \quad n = 1, 2, 3, \dots \quad (3)$$

, where the characteristic Bohr radius a_c decides the spatial extension. Then the analytical tunneling current can be evaluated by the tunneling matrix according to Chen's derivative rule.

$$I_{ana} = \sum_n C_n \left| D \cdot \left(3\kappa^{-2} \frac{\partial^2}{\partial z^2} - 1 \right) \cdot \psi_n(z, d, a_c) + P \cdot \frac{\partial}{\partial z} \psi_n(z, d, a_c) \right|^2 \quad (4)$$

We mainly consider the contribution of $n=1, 2, 3, 4$, so the free parameters are (C_1, C_2, C_3, C_4) , and $(\kappa = \frac{(2m_e\phi)^{\frac{1}{2}}}{\hbar}, d, P, D, a_c)$, nine parameters in total. Using above analytical model and normalizing both the experimental and analytical curves, we obtain the fitting results as follows.

Fig. R2 The analytical fitting. The experimental curves are in red and the fitting ones are in blue. **a-**

f, The fitting of the data presented in Fig. 5d (revised version). **g**, The fitting of the data presented in Fig. 5c. **h**, Corresponding fitting parameters.

As can be seen, it's found that the analytical fitting successfully mimics the shape evolution of the experimental curves and presents reasonable parameters with respect to energy, but the quantitative resemblance is not ideal. After careful reexamination, we attribute it to the reasons listed down below.

1. As mentioned in the next comment of the reviewer, it's unclear whether the $n \geq 4$ Rydberg states are presented in this experiment, so it could be inappropriate to invoke certain series of n in the fitting model.
2. Besides the reasons listed in *Analysis of one-dimensional Rydberg series* in **Methods** section, the moiré potential, interfacial imperfection and inter-exciton interaction may cause the spectral function smearing, leading to complicated spectral weight and wavefunction renormalization. In addition, there is study showing that in two-dimensional system, the Rydberg excitons of different n senses different dielectric environments and thus differ from conventional hydrogenic orbitals [*Exciton binding energy and nonhydrogenic Rydberg series in monolayer WS₂*, Phys. Rev. Lett. **113**, 076802 (2014)]. Therefore, the above model is to some degree oversimplified by treating Rydberg states as isolated and ideal orbital levels.
3. Suffering the microscopic uncontrollability, it's very hard to secure a fully understood tip apex. The apex geometry, structure, and sometimes even the chemical element of apex atom are intangible, along with the difficulties in evaluating the orbital hybridization and spin-orbit entanglement. In this aspect, the model is also oversimplified by neglecting the tip band structure, complicated entanglement, and the contribution from other spherical harmonic contributions such as s , p_x , and d_{xy} , etc. These approximations would cause distortions compared to the realistic situation.
4. Very importantly, Chen's derivative rule is very straightforward in evaluating the tunneling matrix, but by doing this we are neglecting the spatial extension of tip wavefunction (about 1-2 Å for apex orbital of W-tip). Therefore, the analytical current expression is simplified, from the sample wavefunction convoluted with the tip one to the direct derivative of the sample wavefunction. This is an acceptable approximation due to the spatially limited tip wavefunction compared to the sample Rydberg wavefunctions which extend to a long range. However, it would lead to a significant underestimation of the broadening effect caused by the convolution with tip orbital, rendering the features of fitting curves steeper than the experimental ones. As such we think many quantitative inconsistencies can be attributed to the convolution issue.
5. In the experimental I-z spectra, the tip-sample interaction is a big obstacle in reaching an accurate fitting, including both the tip-sample electric field effect and the elastic structural modification caused by atomic force (short-range atomic force and long-range van der Waals and electrostatic force). As the variation of tip-sample distance, the tip-sample interaction is altered, giving rise to elusive stark splitting and elastic relaxation of the apex structure.

To elaborate, they do not visualize any "excitonic complex" in imaging mode but is

inferred from dI/dV plots. It is also not clear the full Rydberg series (up to $n=4$) is seen in the data (Fig. 3).

Reply #2D:

The reviewer is right on both points. The Rydberg charge clouds are profiled in the current spectrum instead of STM scanning images, and to the current step, it's unclear whether the Rydberg levels with higher n ($n=4, 5\dots$) are occupied or not. However, in the imaging mode, it's the lateral two-dimensional distributions of excitons that have been imaged, which show a superlattice structure determined by moiré periodicity. By the word "visualization" we intend to mean that both the in-plane and the out-of-plane charge clouds can be mapped out. To clarify the former point, we have made corresponding modifications in the main text with red fonts:

Abstract, line 16 from the top

...and excitonic in-gap states with Rydberg-like orbitals are directly ~~visualized~~profiled.

Page 3, line 5 from the bottom

...Second, by employing in-depth STM measurements and analysis, we demonstrate the capability of tunneling probe to directly ~~visualize~~profile the charge clouds of interlayer excitonic states.

To clarify the latter point, we make modifications in Page 14, the top paragraph:

*...As the bias is reduced to near the Fermi level, reaching 0.1 V, the E_U is significantly narrowed, and the curve undergoes a gradual transformation into a shape characterized by prominent undulations with two nodes, **indicating a principal quantum number of at least three**. This pattern signifies the predominant contribution of the large n state located in the vicinity of the Fermi level. Thus, it's found in the positive bias range that smaller n states occupy higher energy positions, while larger n states occupy lower energy positions. This result demonstrates that the h -series lie above the Fermi level, of which the n descends with increasing energy. **However, there is an untouchable space above the interface since the tip cannot reach too intimate contact with the sample. Thus, whether Rydberg series with n larger than 3 are occupied and detected or not cannot be surely inferred from the shape of I - z curves.***

Finally, I recommend the authors to add some more high resolution scans of the Moire' lattice at different biases to help the readers understand better.

Reply #2E:

The suggestion is helpful and important. Corresponding data has been made as Extended Data Fig. 6 and presented in the revised version.

Some other minor comments:

1. The choice of the word "negative band gap" is unnecessary in the abstract and a bit misleading.

Reply #2F:

We agree, the word is misleading in the context. We have replaced it and chosen a clear statement in line 3 of the abstract:

...The system of long-lived excitonic states, facilitated by the ~~negative-band gap~~ semimetallic band overlapping, ...

2. Could there be broadening of the V-shaped dI/dV curve due to imperfect interfaces leading to a parabolic shape of the same?

Reply #2G:

We agree. The scattering of disordered interfacial imperfection could renormalize and smear the electronic spectral function, which can be an important contribution to the dI/dV broadening. Therefore, in our opinion, some of the parabolic background of the spectrum may come from the broadened graphite LDOS, and the several humps upon the parabola correspond to the broadened Rydberg levels. We modify corresponding paragraph in the revised version.

Page 6, top paragraph:

...However, in general, all the dI/dV curves of in-gap states exhibit several humps superimposed on a parabola background. Some of the parabolic conductance might be contributed by the V-shape DOS of graphene broadened by interfacial imperfection, but the simple possibility that the electrons purely tunnel from the tip to HOPG substrate through the YbCl₃ insulating barrier is inconsistent with the hump features^{34,35}.

3. Could the authors elaborate on the tip-apex change a bit more quantitatively?

Reply #2H:

To clarify the ambiguity by only saying “tip-apex” change, we modify corresponding paragraph in the revised version.

Page 8, bottom paragraph:

...The tip state can be expressed by the series expansion of spherical harmonic functions in terms of the s, p, d, ... components. Induced by the possible displacement and geometrical crystal field distortion of apex atom, the components of tip state could be altered, resulting in different tunneling matrixes with respect to the sample wavefunction before and after the tip change. If the tip orbital is more sensitive to $|\psi_1\rangle$, that is, the tunneling matrix $M_1 \equiv \langle \psi_1 | U_T | \psi_t \rangle \gg M_2 \equiv \langle \psi_2 | U_T | \psi_t \rangle$ where U_T is the tip potential and ψ_t is tip wavefunction, the iso-current contour mainly contributed by $|\psi_1\rangle$ dominates the height difference, as the orange double-arrowed zigzag line in Fig. 2f indicates the tip trajectory across domain boundary. Once the tip tunneling state is modified into an orbital with $M_2 \gg M_1$, as indicated by the cyan double-arrowed zigzag line, $|\psi_2\rangle$ will dominate the iso-current contour, leading to the reversal of height contrast.

4. Perhaps the authors can also expand on what they mean by "slits" in the sample? Is it

introduced during the growth?

Reply #21:

The reviewer is right. The domains and slits are naturally introduced during the epitaxial growth. We should have provided clear explanation for this. Now it has been revised.

Page 6, in the middle:

...Different orientational domains of YbCl_3 monolayer are naturally introduced during the growth, and form a slit at the domain boundary. The chosen sample area is representative with two neighboring domains separated by a slit.

Reviewers' comments:

Reviewer #1 (Remarks to the Author):

With the reply from Zhao et al., I'm still not fully convinced by their arguments. The wavefunction of the Rydberg states should inherit the 4f orbital nature of YbCl₃ and I don't think it can have such profound z extensions. Meanwhile, the authors didn't demonstrate the charge neutral character of the so-called Rydberg excitonic states. STM is a charge sensitive measurement and cannot probe charge-neutral excitation states as these states should be charge-incompressible and exciton-compressible. Or in other words the peaks in dI/dV shouldn't correspond to excitonic states but charge states. I hence cannot suggest publication of this work. Though I cannot provide better explanations of the z modulation of dI/dV at this stage, I suggest the authors to carefully examine their measurement with different tips and on different regions of the sample (e.g. on bare HOPG). If the observation is strongly reproducible, and for any future versions of the work submitting elsewhere, I recommend toning down the claim of "interlayer Rydberg exciton complex" in the title to something like "Rydberg-like states".

Reviewer #2 (Remarks to the Author):

The authors have addressed my questions to my satisfaction. I can recommend publication to Nature Communications.

According to the latest comments of reviewer #1, it seems that the three main concerns in the last round about the charge-transfer process, exciton ground state, and moiré singularity have been resolved. Here, regarding the extra concerns appearing in the latest comment, we answer point by point as follows.

Response to the reviewer #1

With the reply from Zhao et al., I'm still not fully convinced by their arguments. The wavefunction of the Rydberg states should inherit the 4f orbital nature of YbCl₃ and I don't think it can have such profound z extensions.

Reply:

This issue has already been discussed in the Methods section which is listed later. The key point is the transferred electron filled into Yb-4f band feels the strong screening of the initial Yb-4f¹³ valence electrons, rendering a much larger wavefunction. Different from the inherent 4f electrons with highly localized orbitals, the electron transferred into YbCl₃ is a doped electron subject to a charge-neutral background. Such a situation is the same as the **shallow donors** in semiconductors where the doped electron is bound by the screened Coulomb potential of the donor ions which leads to hydrogen-like orbitals of donor level with large spatial extension (**envelope wave functions**) and **Rydberg series** [chapter 4, Peter Y U, Cardona M. *Fundamentals of semiconductors: physics and materials properties*. Springer Science & Business Media, (2010).].

“Analysis of one-dimensional Rydberg series” paragraphs of the Methods section, Page 19, bottom line in the latest revised manuscript:

...(ii) From the orbital perspective, on the one hand, the transferred e and h populate the YbCl₃ conduction band and graphene valence band respectively, they are therefore imparted with the features of Yb-4f and C-2p orbital. This means that besides the mutual Coulomb bound potential between e and h, they are also subjected to the ionic potential and submit to the atomic orbital physics, which imposes huge influence on the real-space character of exciton states, especially noting that the Yb-4f orbitals are extremely localized and highly correlated with heavy effective mass and strong spin-orbit coupling. On the other hand, acting as extra charges embedded into an almost electroneutral crystalline background, the transferred e and h feel much mitigated Yb- and C- ionic potential because the initial valence electrons screen the ion cores, producing difficulties for assessing the exact orbital influence on wavefunction.

We modified the main text to clarify this point:

Page 11, second paragraph,

“Besides the energy-level diagram of charge-transfer excitons, their few-body wavefunctions near the interface are rather complicated as the transferred e and h mutually bind and screen each other in an interactive manner. Especially considering the 4f orbital character inherited by the transferred e, the e-states should be highly localized. However, subjected to an almost charge-neutral lattice background, the doped electron filled into the Yb-4f band feels the strong screening of Yb ionic potential by the initial Yb-4f¹³ valence electrons, rendering the Coulomb attraction from adjacent

doped h a comparatively dominant interaction and a large Rydberg wavefunction of transferred e, which is reminiscent of the shallow donors in semiconductor with large envelope wavefunction and Rydberg series³⁹.”

...Meanwhile, the authors didn't demonstrate the charge neutral character of the so-called Rydberg excitonic states. STM is a charge sensitive measurement and cannot probe charge-neutral excitation states as these states should be charge-incompressible and exciton-compressible. Or in other words the peaks in dI/dV shouldn't correspond to excitonic states but charge states.

Reply:

This issue has already been elaborated in the “Tunneling accessibility towards charges of exciton complex” paragraphs of the Method section. Rather than a charge-neutral elementary particle, the exciton is composite boson formed by the bound state of opposite charges. In other words, **it is the electron or hole levels within the electron-hole bound states that are separately probed by STM**. The electrons trapped by their counterpart holes carry charges that can be accessed via STM. As a typical example, the **image potential state at metallic surface is a bound electron-hole state which can be readily accessed by STM** [[Niesner, Daniel, and Thomas Fauster. *Image-potential states and work function of graphene*, Journal of Physics: Condensed Matter 26.39: 393001 (2014)].

We modified the main text to clarify this point:

Abstract, middle part:

“...Here, without the aforementioned approaches to maintain excitons, we instead realize the ground-state interlayer exciton complexes through the intrinsic charge transfer in monolayer YbCl₃/highly oriented pyrolytic graphite (HOPG) heterostructure, and probe the charge clouds of electron and hole within the exciton separately.”

Page 4, line 1 from top:

*“...Owing to the extreme spatial and energy resolution provided by tunneling probe, we reveal the excitonic energy diagram, and map out the three-dimensional distributions of Rydberg ~~states~~ **electron- and hole-**states including both the out-of-plane nodal structure and moiré-dominated in-plane periodic arrangements.”*

Page 11, first paragraph, the last sentence:

*“...For both cases, in increasing order of energy, the principal quantum number n of e-series is ascending while that of h-series is descending. **It has been demonstrated in the STM studies of image potential states that the Rydberg e-levels form e-h bound states are probed and manifested as discrete tunneling channels³⁸.”***

Page 12, line 8 from bottom

*“Next, we demonstrate that the out-of-plane nodal structure of Rydberg charge densities of **e- and h-states** can be profiled by the tunneling probe....”*

Page 14, line 9 from the top:

“...This result demonstrates that it is the *h*-Rydberg levels lying above the Fermi level that are probed at positive bias, as their principal quantum number *n* descends with increasing energy.”

...Though I cannot provide better explanations of the *z* modulation of dI/dV at this stage, I suggest the authors to carefully examine their measurement with different tips and on different regions of the sample (e.g. on bare HOPG). If the observation is strongly reproducible, and for any future versions of the work submitting elsewhere, I recommend toning down the claim of “interlayer Rydberg exciton complex” in the title to something like “Rydberg-like states” .

Reply:

We have spent much time and effort to secure the data reproducibility before writing the manuscript. The reexamination experiments mentioned by the reviewer have already been done, and relevant data from our inventory is shown as follows. All the *I*-*z* curves of many tries on YbCl_3 (different tips, different sample regions) show the same oscillatory behavior, and all the *I*-*z* curves taken either from the bare HOPG region of the YbCl_3 -HOPG sample, pure HOPG, or the monolayer halides like FeCl_2 , CoCl_2 (samples we studied before) show the same exponentially decaying feature.

Fig. **a-c** I-z measurements performed on different regions of monolayer YbCl_3 with different tips. **d** I-z curves taken on the monolayer CoCl_2 on HOPG. **e** I-z curves taken on the monolayer FeCl_2 on HOPG, which is extracted from our previous publication [*Nanoscale* **12**, 16041 (2020)], where the bias-dependent behavior is explained. **f** I-z curves obtain on HOPG.

Not only the reviewer cannot provide a better explanation, but also we go over a rich variety of possibilities to find the answer to this novel observation. The charge-transfer exciton is a brand-new paradigmatic discovery, and to confirm this we have excluded a rich variety of trivial explanations and artifacts.

We have included above data as Extended Fig. 7. We hope our reply and revision can address the concerns.

REVIEWERS' COMMENTS

Reviewer #1 (Remarks to the Author):

Given the new set of data (Extended Fig.7) and discussions by the authors, the manuscript now looks more complete and convincing in arguing about the presence of Rydberg-like states. The discovery is also novel and can provide insights for a general understanding of interfacial effects between different 2D materials. I would like to suggest acceptance of the revised manuscript in the current form.